# Thermal Effects of Natural Gas and Syngas Co-Firing System on Heat Treatment Process in the Preheating Furnace

**Piotr Jóźwiak [1,2,]\*** , **Jarosław Hercog [1]**, **Aleksandra Kiedrzyńska [1]** , **Krzysztof Badyda [2]** and **Daniela Olevano [3]**

1 Thermal Processes Department, Institute of Power Engineering, 01-330 Warsaw, Poland; jaroslaw.hercog@ien.com.pl (J.H.); aleksandra.kiedrzynska@ien.com.pl (A.K.)
2 Institute of Heat Engineering, Warsaw University of Technology, 00-665 Warsaw, Poland; badyda@itc.pw.edu.pl
3 Centro Sviluppo Materiali S.p.A., 00128 Rome, Italy; daniela.olevano@rina.org
\* Correspondence: piotr.jozwiak@ien.com.pl; Tel.: +48-22-3451-407

**Abstract:** Preheating furnaces, which are commonly used in many production sectors (e.g., iron and steel), are simultaneously one of the most energy-intensive devices used in the industry. Partial replacement of natural gas with biomass-derived synthesis gas as a fuel used for heating would be an important step towards limiting industrial $CO_2$ emissions. The time dependent computational fluid dynamics (CFD) model of an exemplary furnace was created to evaluate whether it is possible to obtain 40% of energy from syngas combustion without deterioration of thermal parameters of the treated load. As an outcome, a promising method to organize co-firing in the furnace was indicated. The obtained results show that the co-firing method (up to 40% thermal natural gas replacement with syngas), assuming low air-to-fuel equivalence ratio ($\lambda_{NG}$ = 2.0) and even distribution of power among the furnace corners, lead to satisfactory efficiency of the heat treatment process—the heat transferred to the load exceeds 95% of the heat delivered to the load in the reference case), while carbon dioxide emission is reduced from 285.5 to 171.3 kg $CO_2$/h. This study showed that it is feasible (from the heat transfer point of view) to decrease the environmental impact of the process industries by the use of renewable fuels.

**Keywords:** CFD modelling; heat treatment process; industrial furnaces; natural gas substitution; syngas co-firing

## 1. Introduction

Preheating furnaces use substantial amounts of energy in industrialized countries. A great number of furnaces and substantial energy consumption accompanying the heat treatment processes lead to the significant carbon footprint in Europe and beyond. In Germany, industrial processes are responsible for nearly 40% of natural gas consumption [1]. In Poland, over 45% of natural gas is used by industry and construction sectors [2]. High energy consumption of furnaces results from the need of heating the processed load to a specific temperature over a certain period of time. On an industrial scale, where the volume of the treated material is large, it means high demand for thermal power, associated with proportional operating cost and environmental impact.

On the other hand, it is deemed necessary to reduce the anthropogenic pressure on climate change [3] and the diversity of the biosphere, correlated with the $CO_2$ emissions from combusting fossil fuels. This is followed by concrete actions at the level of the European Union: 2020 climate and energy package, 2030 climate and energy framework, 2050 long-term strategy, and the resulting

initiatives for public-private partnerships like Sustainable Process Industry through Resource and Energy Efficiency aiming to make the European process industry more efficient and sustainable, and less resource consuming [4].

Natural gas is a commonly used energy source in the industry due to its wide applicability and availability at a low price. Although its $CO_2$ emission factor is the lowest of all hydrocarbons and it is considered as a much cleaner alternative to oil or coal, combustion of natural gas is still associated with polluting the atmosphere with greenhouse gases and the climatic consequences of that. Thus, measures limiting both fossil fuel consumption and gaseous emissions from the industry are sought. In the steel industry, the process by-products such as coke oven gas (COG), blast furnace gas (BFG) [5], and basic oxygen furnace gas (BOFG) can be used as a feedstock for reheating furnaces [6]. Usage of those gases in reheating furnaces and annealing lines with radiant tube burners was briefly analyzed [7]. It was found that there are no significant constraints for application of those gases, although cleaning the gas has to be done prior usage to avoid damaging the equipment. Researchers [8] investigated the influence of the syngas impurities on scale formation on steel slabs in the reheating furnaces, and found that they can cause corrosion and slagging.

Usage of other alternative gases (such as biogenic syngas fuels) has not been extensively explored, especially in the preheating furnaces where the temperatures are significantly lower and the conditions for complete combustion are not favorable. Moreover, the biogenic syngas may contain a significant amount of tars, which can either require gas cleaning (inherently connected with exergy losses) or cause difficulties in process equipment selection and operation in temperatures above the tars dew point.

Other contaminants which could be present in the biogenic syngas are halogen species (Cl, Br) [9] and, to a smaller extent, the alkali metals (Na, K) [10] which can cause operational problems [11]. Taking into account the above mentioned, there are no studies of the application of the syngas coming from the biomass gasification in the preheating furnaces.

One of the possible methods for low-emission combustion is partial substitution of natural gas with gaseous biofuels, whose $CO_2$ emission factor, according to the European Commission regulations, is equal to 0 [12]. An example of such gas is syngas obtained from the solid biomass gasification process.

There is a noticeable interest in the issue of replacing natural gas as a source of heat in the industry with renewable gas fuels. The possibility of reducing $CO_2$ emissions by co-firing syngas in gas turbines [13] and coal-fired boilers [14] has been studied. Research has been done on emission levels of nitrogen oxides (NOx), carbon monoxide (CO), polycyclic aromatic hydrocarbons (PAH), and volatile organic compounds (VOC) accompanying the combustion of syngas [15]. Partial replacement of natural gas by biogas has been proven to be a sensible approach for implementation in the glass processing industry as no negative effects in product quality have been observed [16]. There have been no studies investigating the impact of hot biogenic syngas co-firing on heat treatment process parameters in steel preheating furnaces. The preliminary study preceding this work has shown that the use of low-calorific alternative fuels (specifically: basic oxygen furnace gas and biogenic synthesis gas) does not excessively change the course of the heating process, and that the introduction of dedicated syngas burners is a preferred option of organizing co-firing in the considered furnace [17].

The goal of this study is to analyze the effects of partial substitution of natural gas with biomass-derived synthesis gas on temperature of a typical load treated in an exemplary steel sector preheating furnace. The potential of introducing carbon-neutral low-calorific syngas to the process sector has not yet been studied. The results are to be compared with the ones gathered from the furnace fired in standard operation mode (leaning on natural gas only) in order to assess the possibility of applying this method of carbon dioxide emission reduction to other industrial units without worsening thermal parameters of the process.

## 2. Materials and Methods

*2.1. Model Setup*

2.1.1. Furnace

The numerical model is based on the existing preheating furnace BOSIO 1 located in Store, Slovenia. For this study a medium-size preheating furnace was selected. This particular furnace was chosen because it is a representative example of thousands of similar, simple operation units installed around the world. The characteristic batch loading (using bogie hearth) enables high flexibility for thermal treatment of loads of different sizes and shapes. The obtained results should be extrapolated with ease to furnaces of similar type.

This gas-fired unit is equipped with four burners combusting natural gas (NG) able to operate with power up to 400 kW each, four independent recuperators preheating the air from room temperature to 60 °C (333 K), and a doubled chimney with the natural draft. During the heat treatment a vortex-like flow structure inside the furnace is created, and direct contact between streams of flue gas and the load is avoided (Figure A1). For co-firing purposes, an addition of two burners dedicated to combust syngas (SG) was proposed—their operating power is 360 kW each, and they are located in the vicinity of NG burners to ensure complete combustion of syngas. The geometry of the furnace model is presented in Figure 1. The load is a 4.1 m long steel cylindrical mold with an outer diameter of 1.8 m. It was assumed that its material density is 7700 kg/m$^3$, specific heat is 502.5 J/(kg·K), and thermal conductivity is 50 W/(m·K).

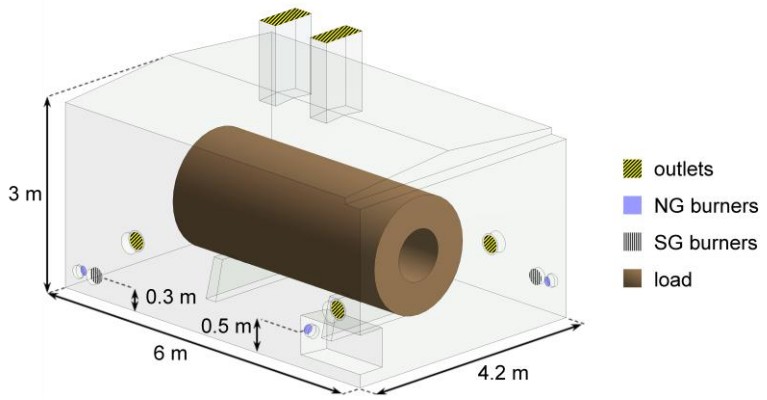

**Figure 1.** Geometry of the furnace model (NG—natural gas, SG—synthesis gas).

Numerical domain of the furnace was discretized with a hybrid mesh—structured elements form the load, chimneys, and part of the syngas burners interior, and the rest of the geometry was filled with polygonal elements (total number of cells: 875,119).

Figure 2 shows the NG and SG burners' primary (17% of comburent air, marked blue) and secondary (83% of burner air, marked red) preheated air inlets separated by the inlet of the respective fuel (green for syngas, yellow for natural gas). The secondary air swirl angle is 30°. The air consists of 77.45% nitrogen, 20.59% oxygen, and 1.96% water vapor by volume, what is an exemplary value corresponding to air at 85% relative humidity at 20 °C. The heat leaves the furnace to ambient at 20 °C (293 K) through convection: heat transfer coefficient at the outer walls of the furnace was specified as 25 W/m$^2$·K, and with exhaust gas through the chimney outlets. Thermal insulance of the walls was estimated as 0.1 m$^2$·K/W (0.2 m$^2$·K/W for the walls surrounding NG burner no. 1 (the closest to the viewer in Figure 1) due to their reduced thickness of 0.1 m). Emissivity was 0.9, regardless of the temperature level and the material type.

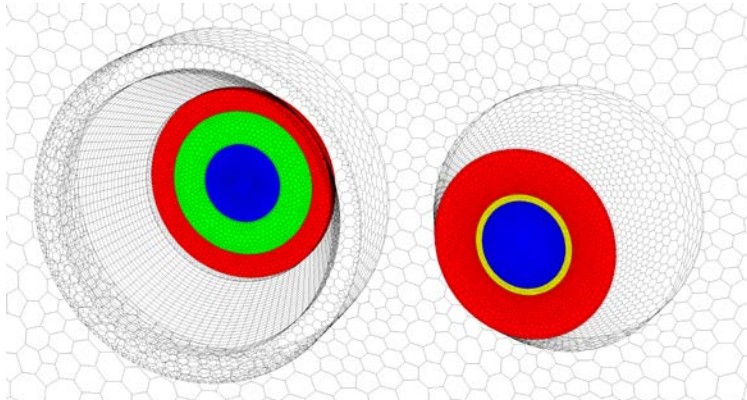

**Figure 2.** Discretization of surfaces near syngas (left) and natural gas (right) burners—colors mark respective air and fuel inlets (blue—core air, yellow—NG, green—SG, red—secondary air).

Total power of the considered furnace is limited to 1440 kW. The same power was fed in fuel in the numerical model. The influence of the recuperators extracting a fraction of the exhaust gases from the inside of the furnace on the gas flow was represented by assigning fixed velocity (1.33 m/s) to the recuperator outlets. In cases where no syngas is fed into the system, inlets of the SG burners were treated as adiabatic walls.

### 2.1.2. Fuels

The chemical composition of both considered fuels was specified by the furnace operator. Natural gas available in the facility contains the following combustible species: 97.97% methane, 0.76% ethane, and 0.38% other hydrocarbons (by volume)—the rest consists of carbon dioxide, nitrogen, and oxygen (<0.1%). NG is fed to the furnace at 20 °C (293 K).

The alternative fuel is generated by the located on-site fluidized bed gasifier, in which air is used as a gasification agent. The gasifier produces up to 1 MW power in gaseous fuel (ca. 0.7 MW on average). The lower heating value of a typical syngas produced by this type of gasifiers fluctuates in the range of 3–7 MJ/m$^3_N$ [18]. As one of the adopted objectives to achieve higher efficiency of the co-firing system is to preserve the initial high temperature of syngas, it was assumed that the SG produced by the gasifier is not cooled down on the way to the furnace, and it enters the dedicated burners at 600 °C (873 K). Hot syngas is not cleaned from tars—SG temperature should not drop below 350 °C (633 K) to avoid tar condensation and contamination of the installation. The amount of energy contained in tars (below 0.04 MJ/m$^3_N$ of SG, as tars concentration in the considered syngas is lower than 1 g/m$^3_N$) is insignificant for the course of the heat treatment.

Presence of species with concentrations lower than 0.5% vol. was neglected. The composition of the gasification product is variable over time and estimated average values were chosen for the calculations (Table 1). Lower heating values for natural gas and synthesis gas are 50.05 and 4.52 MJ/kg, respectively. The fuels were considered in a dry state.

**Table 1.** Chemical composition of natural gas (NG) and syngas (SG) adopted for simulations.

| Component | Natural Gas (% vol.) | Syngas (% vol.) |
|---|---|---|
| $O_2$ | 0 | 1 |
| $CO_2$ | 0 | 12 |
| CO | 0 | 21 |
| $CH_4$ | 98 | 3 |
| $C_2H_6$ | 2 | 0 |
| $H_2$ | 0 | 14 |
| $N_2$ | 0 | 49 |
| $H_2O$ | 0 | 0 |

Despite the content of $CH_4$, CO, and $CO_2$, syngas is regarded as carbon-neutral because of its biomass origin. This means carbon dioxide emission factor for SG is 0 kg $CO_2$/GJ, instead of 135 kg $CO_2$/GJ as its gas composition would indicate. NG emission factor is 55 kg $CO_2$/GJ.

*2.2. Operation Modes*

To evaluate the influence of NG-SG co-firing on the quality of the heat treatment, nine different furnace powering scenarios were developed and implemented in the furnace model—the test matrix showing the case numeration and respective substitution rates, NG burner power, and air supply levels is shown in Figure 3. For each case the furnace power is equal to the initial one, i.e., 1440 kW. Three main factors that can affect the process course were identified and tested:

- Mode of work (NG only or co-firing)—to value the impact of SG presence, e.g., two additional flames, on the heat treatment;
- Air–fuel equivalence ratio at natural gas burners $\lambda_{NG}$ (2.0, 3.0, or resulting from feeding 1980 $m^3_N$/h (at normal conditions ca. 0.71 kg/s) of air to the natural gas burners, i.e., the amount supplied in the original BOSIO 1 furnace—to check the correlation between the amount of air fed to the furnace, the resulting change in gas motion, and the heat transfer;
- Power distribution among the corners of the furnace (even or uneven)—to determine the effect of balancing the power outcome of the NG-SG burner pair and the two NG burners without adjacent SG burner, accomplished by adjusting the power of NG burners so at every corner of the furnace 360 kW of heat is released.

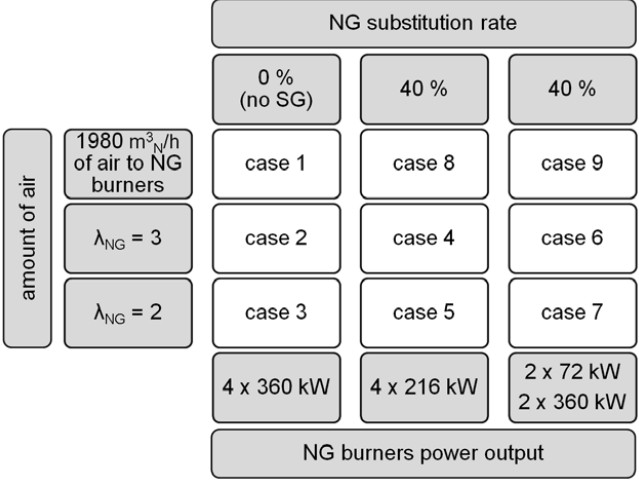

**Figure 3.** Parameters of the considered furnace operation modes.

The air–fuel equivalence ratio for SG burners $\lambda_{SG}$ is fixed to 1.15, because syngas as a low calorific fuel is considered more prone to incomplete combustion or flame blowout [19], and as such needs conducive conditions to avoid it.

The first 2 h of the process were simulated—during that time the burners are turned on continuously, and the tracked indicators, i.e., average temperature and maximum temperature difference of the load are expected to increase. At the beginning of the heat treatment (t = 0 s), the load and the air inside the furnace are 20 °C (293 K). Case 1 (no syngas, 1980 $m^3_N$/h of air) serves as the reference case.

### 2.3. Calculation Method

CFD tools are widely used to simulate and optimize the processes of heat transfer [20] and energy release from various fuels [21]. A transient 3D model was prepared and used to determine the changing conditions inside the furnace over time. The model based on URANS approach covers turbulent gas flow by solving three-dimensional Reynolds-Averaged Navier-Stokes equations with realizable k-ε turbulence model applied to solve the Reynolds stresses. Gravitational forces were included. The solver is pressure-based and because adaptive time step sizes (0.1 s for the first 1 s of the process, 1 s for the period between 1 s and 15 s of the process, and 15 s for the rest of the process) were used in the simulations, the coupled algorithm for calculating pressure was chosen.

The gas is a multicomponent single-phase mixture composed of $CO_2$, $CO$, $CH_4$, $C_2H_6$, $H_2$, $O_2$, $H_2O$, and $N_2$, whose local mass fractions are predicted through solving the conservation equations for each species (except for nitrogen, which is the balancing species), and both density (following the incompressible ideal gas law) and specific heat of each compound are temperature dependent. Species can participate in the volumetric chemical reactions Equations (1)–(4), which are the source of thermal energy in the system.

$$C_2H_6 + 2.5\,O_2 \rightarrow 2\,CO + 3\,H_2O, \tag{1}$$

$$CH_4 + 1.5\,O_2 \rightarrow CO + 2\,H_2O, \tag{2}$$

$$CO + 0.5\,O_2 \rightarrow CO_2, \tag{3}$$

$$H_2 + 0.5\,O_2 \rightarrow H_2O \tag{4}$$

The rate of these chemical reactions is controlled by turbulent mixing, that is an acknowledged approach used for modelling non-premixed combustion (eddy-dissipation model) [22].

The numerical model allows the heat to be transferred by convection, conduction (within the load's volume), and radiation mechanisms. Radiative heat transfer is calculated using discrete ordinates method with the weighted-sum-of-grey-gases model included. Temperature is obtained through solving the energy equation. The CFD simulations were performed in Ansys Fluent 19.0.

### 2.4. Discretization Error

In order to evaluate the accuracy of the numerical model, the influence of the meshing method on the results was evaluated. Ten different discretization grids were analyzed—all of them consist of structured and polygonal elements. Two examples of the tested meshes are presented in Figure 4.

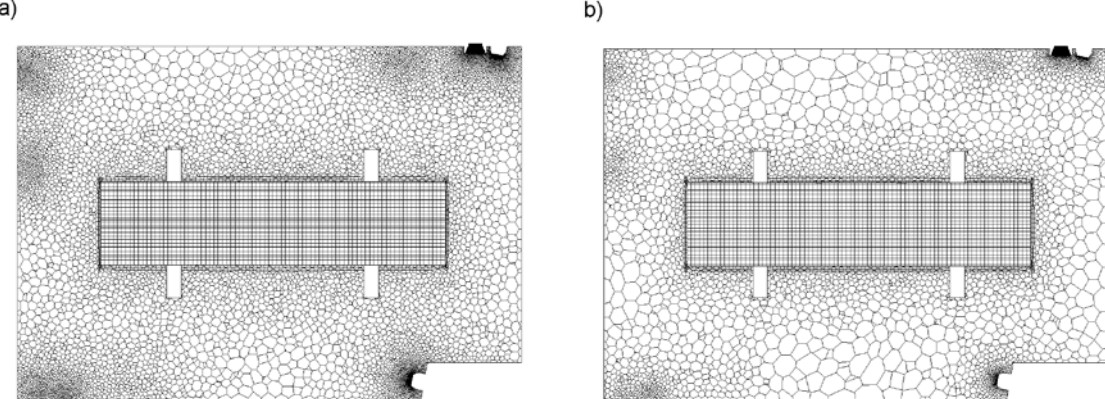

**Figure 4.** Examples of discretization mesh located 0.5 m above the furnace bogie level consisting of (**a**) 875,119 and (**b**) 533,386 elements.

The volume average temperature of the load (ALT) was chosen for an indicator of the discretization error. Case 1 was simulated using the same model settings and the same time step sizes, and the obtained results of ALT after 2 h of the process were compared (Figure 5). The black point on the chart marks the mesh used for performing the calculations for cases 1–9.

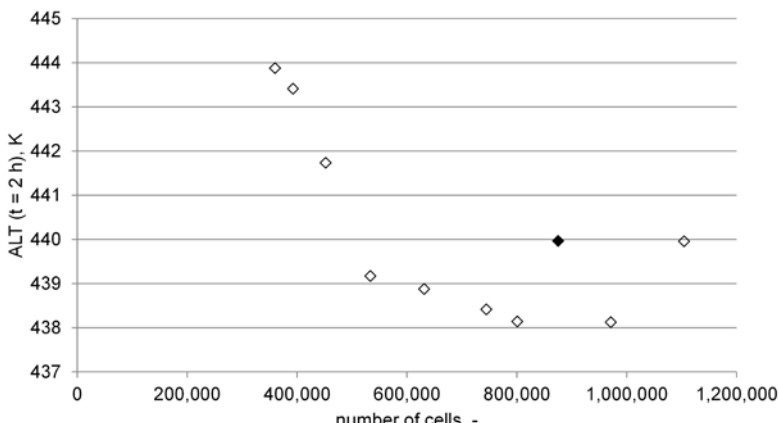

**Figure 5.** Convergence study results—volume average load temperature after 2 h of the heat treatment for different discretization meshes.

One can see that enhancing the grid resolution keeps the values of ALT within the 2 K range, what appears to be an acceptable value of error, i.e., less than 2% when compared to the value change of average load temperature ($\Delta$ALT).

## 3. Results

The results of the numerical calculations contain information about, among others, temperature field in the gas and solid domains. Exemplary visualization of the obtained temperature results for case 7 is shown in Figures 6 and 7.

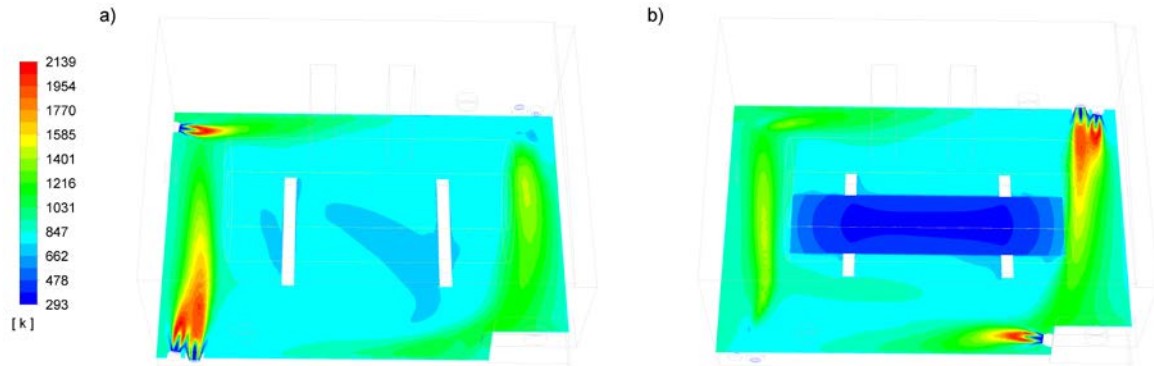

**Figure 6.** Temperature contours after 2 h of the heat treatment (**a**) 0.3 m and (**b**) 0.5 m above the furnace bogie level—case 7.

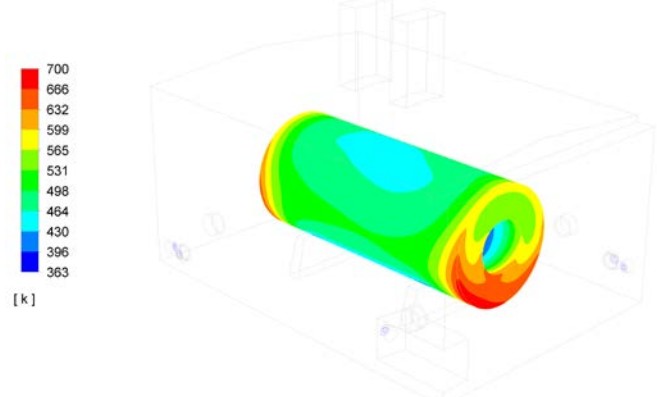

**Figure 7.** Temperature of the load surface after 2 h of the heat treatment—case 7.

In this study the focus was on the mold temperature levels, especially after 2 h of the treatment. Acquisition of the data took place at every time step, therefore at least every 15 s of the simulated process time. The results for the two analyzed cases (1—the reference one, and 7—the SG co-firing scenario with the results most similar to case 1) are presented in Figure 8.

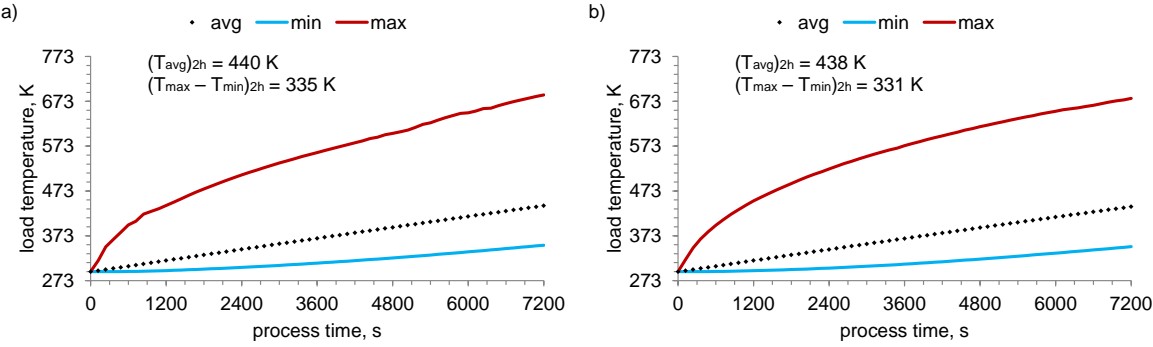

**Figure 8.** Volume average (avg), minimum (min), and maximum (max) temperatures of the load during the first 2 h of the heat treatment: (**a**) case 1, (**b**) case 7.

The applicability and efficiency of each powering mode was assessed on the basis of load temperature difference (LTD) and volume average load temperature (ALT), respectively. Value of LTD is defined as the temperature difference between the hottest ($T_{max}$) and the coldest ($T_{min}$) point of the load—lower LTD means better thermal uniformity. ALT is an indicator of thermal efficiency, i.e., the

ratio of the energy fed in fuel to the energy received by the load—higher values of this measure denote improved usage of heat.

Substituting 40% of NG by low-calorific SG with a constant amount of air supplied to the NG burners has a negative or no effect on thermal uniformity of the load, depending on whether the power among the furnace corners has not or has been equated, respectively (cases 8, 9; Figure 9a). Introducing syngas, while maintaining $\lambda_{NG}$ at 2.0, increases the maximum load temperature difference up to the level noted in the reference case (cases 5, 7; Figure 9a), for which $\lambda_{NG}$ is 1.42. Adjusting the natural gas burners' power in cases where $\lambda_{NG}$ is equal to 2.37 or 3.0 is noticeably beneficial from the LTD perspective (cases 4, 6, 8, and 9).

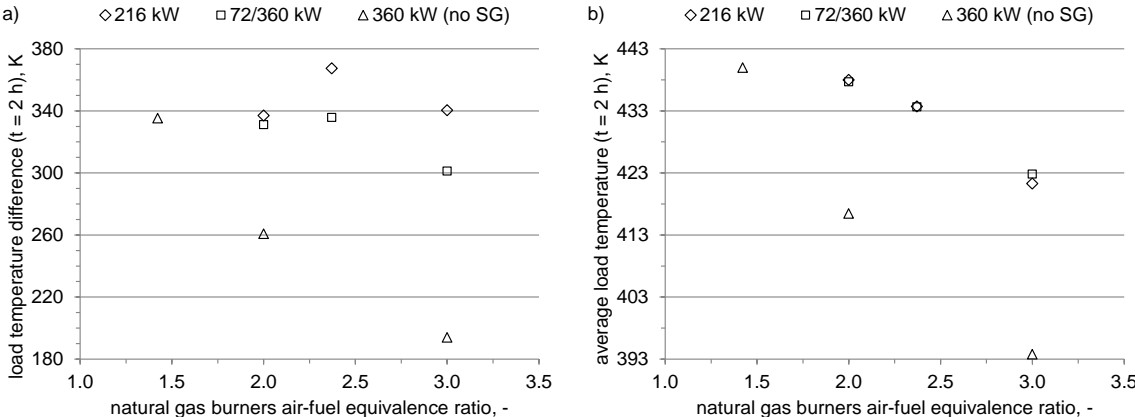

**Figure 9.** Load temperature difference (**a**) and average load temperature (**b**) values for different ways of supplying power (number of kW represents the power of NG burners).

Use of different NG burners power outputs has no impact on average load temperature, regardless of $\lambda_{NG}$ applied (cases 4, 6 and 5, 7 and 8, 9; Figure 9b). Reducing the air–fuel equivalence ratio is correlated with elevation of the heating efficiency, especially in case of no syngas addition. Implementation of co-firing significantly raises the amount of heat absorbed by the load—for $\lambda_{NG}$ equal 3.0 by 28.1% ± 0.7%, and for $\lambda_{NG}$ equal 2.0 by 17.3% ± 0.1%, reaching 98.5% ± 0.2% (cases 5 and 7) of the reference case heating efficiency.

Within 1 h the furnace powered conventionally (cases 1–3) consumes ca. 103.6 kg of natural gas and emits 285.5 kg of $CO_2$ (under the assumption of complete fuel combustion). When 40% of NG is replaced by SG (cases 4–9), the amount of fossil carbon dioxide added to the atmosphere drops by 40% as well—in that case the emission is 171.3 kg $CO_2$/h.

## 4. Discussion

The results of numerical simulations for the considered cases (Figure 3, Section 2.2) are compared in Figure 10. The value of change of average load temperature (ΔALT), calculated as the difference between ALT after 2 h of the process and the initial temperature of the system (i.e., 20 °C), is proportional to the amount of energy transferred to the load. ΔALT for case 1 serves as the reference value.

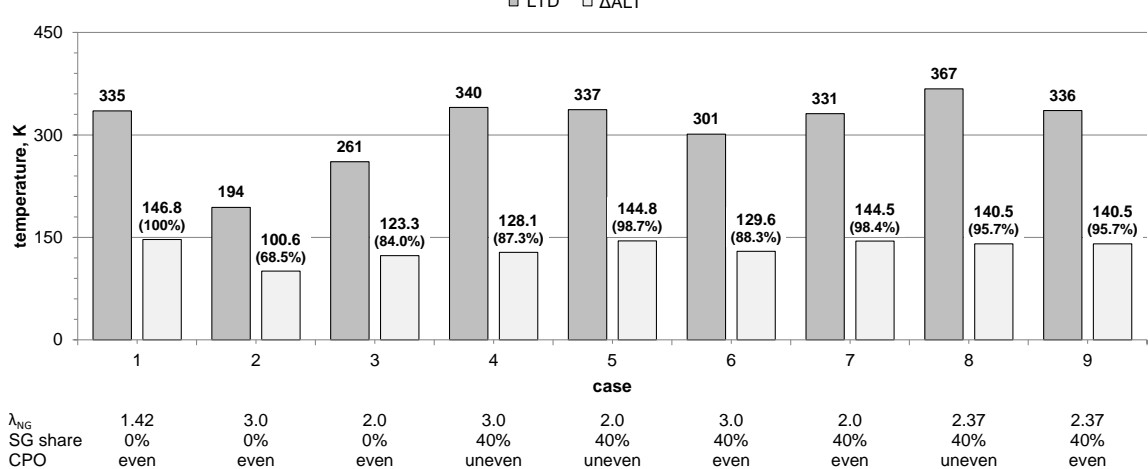

**Figure 10.** Load temperature difference (LTD) and change of average load temperature from initial temperature (ΔALT) after 2 h of the heat treatment (CPO—furnace corner power output).

Conducted CFD calculations show that equalization of the power output in each of the corners has a positive effect on the load temperature difference, especially in case of increased air–fuel equivalence ratio for natural gas burners and the resulting reduction of exhaust gas temperature. Replacing natural gas with biogenic syngas in the tested co-firing setup proved to achieve satisfactory efficiency of the heat treatment process—in four out of six analyzed co-firing scenarios the amount of heat transferred to the load exceeds 95% of the heat delivered to the load in the reference case (case 1). This effect is correlated with the lower SG flame temperature (Figure 6), and the fact that the streams of hot flue gases do not directly hit the load.

Analysis of the simulations results revealed a promising way of substituting 40% of natural gas in the preheating furnace with the considered biomass-derived renewable fuel, which leads to significant reduction of $CO_2$ emissions, thus, smaller environmental impact. This is especially prominent in case 7, in which natural gas burners operate at $\lambda_{NG} = 2.0$ on two power levels: 72 kW (the ones near the syngas burners) and 360 kW (the other two burners), so in each of the furnace corners 360 kW of heat is generated. An important advantage of case 7 over cases 9 and 6 (especially the latter one) is that lower air-to-fuel ratio at the NG burners prevents the creation of strong stream of flue gas flowing out of natural gas burners operating at 360 kW (Figure A6), that could potentially disturb the combustion in the SG burners, which are inclined to less stable operation. Aggravation of this effect, caused by enhancing the stream of flue gas leaving the NG burners can also be noticed by comparing the gas flow patterns available in the Appendix A for the following sequences of cases: 1-3-2, 7-9-6, and 5-8-4 (Figures A1–A9).

Simultaneously, the temperature results show that the co-firing method assuming low air-to-fuel equivalence ratio and uneven distribution of power among the furnace corners (corner without SG burner: 216 kW, corner with SG burner: 504 kW) can still lead to satisfactory results (case 5). Analysis of the streamlines and temperature profiles for cases 5 and 7 shows that power equalizing on the one hand reduces the risk of local load overheating (Figure A5), but on the other hand promotes creation of strong jet-like flows, which hit and locally heat up inner walls of the furnace (Figure A7). It is likely that the optimal power balance between the furnace corners, where the scale of both phenomena is reduced and the efficiency achieves its maximum, lies between the analyzed values.

Based on these results, further work to evaluate the possibility of partial replacing natural gas consumption with alternative fuels (e.g., biomass-derived gaseous fuels or off-gases) can be done, especially for other types of furnaces (e.g., the ones continuously heating and melting stream of material). It is possible that in some cases, depending on the limits on LTD values, it may be necessary

to increase the air–fuel equivalence ratio for syngas burners in order to lower SG flame temperature and improve thermal uniformity of the load.

**Author Contributions:** Conceptualization, P.J., A.K. and J.H.; methodology, P.J.; validation, P.J.; formal analysis, P.J.; investigation, P.J.; writing—original draft preparation, P.J.; writing—review and editing, P.J., J.H., A.K., K.B. and D.O.; visualization, P.J.; supervision, J.H.; project administration, J.H.; All authors have read and agreed to the published version of the manuscript.

**Funding:** This research was funded by European Union's Horizon 2020 research and innovation programme, grant number 723803. The APC was funded by the Institute of Power Engineering.

**Acknowledgments:** The authors would like to thank Jernej Mele (CPPE d.o.o.) and Matej Drobne (Valji d.o.o.) for support and providing process data.

**Conflicts of Interest:** The authors declare no conflict of interest.

## Appendix A

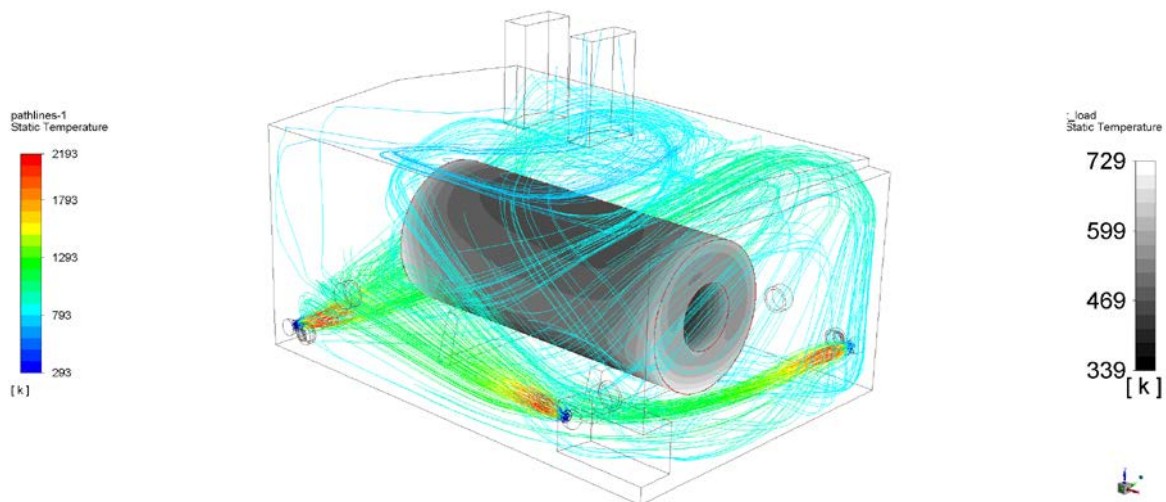

**Figure A1.** Gas path lines and load surface temperature after 2 h of the heat treatment—case 1.

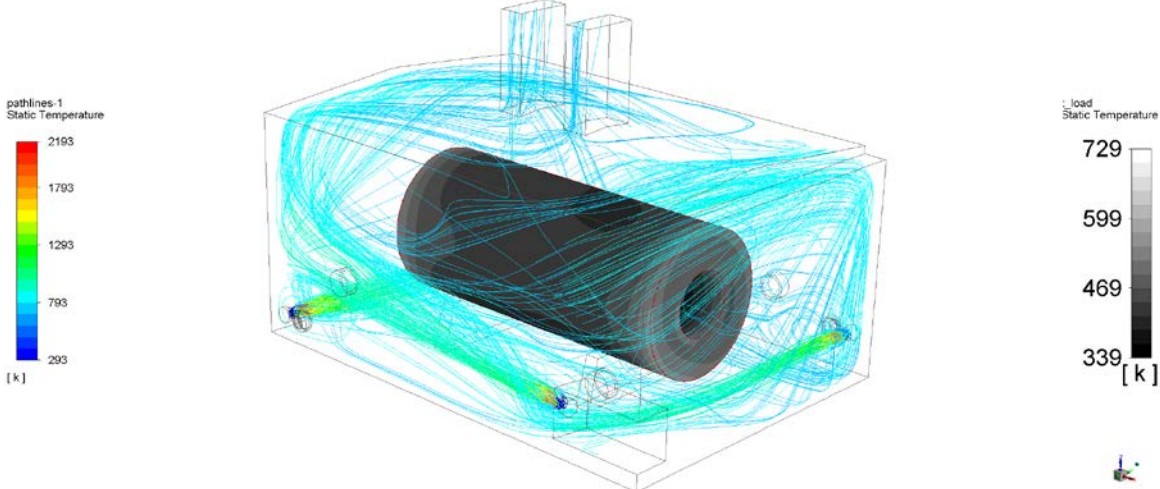

**Figure A2.** Gas path lines and load surface temperature after 2 h of the heat treatment—case 2.

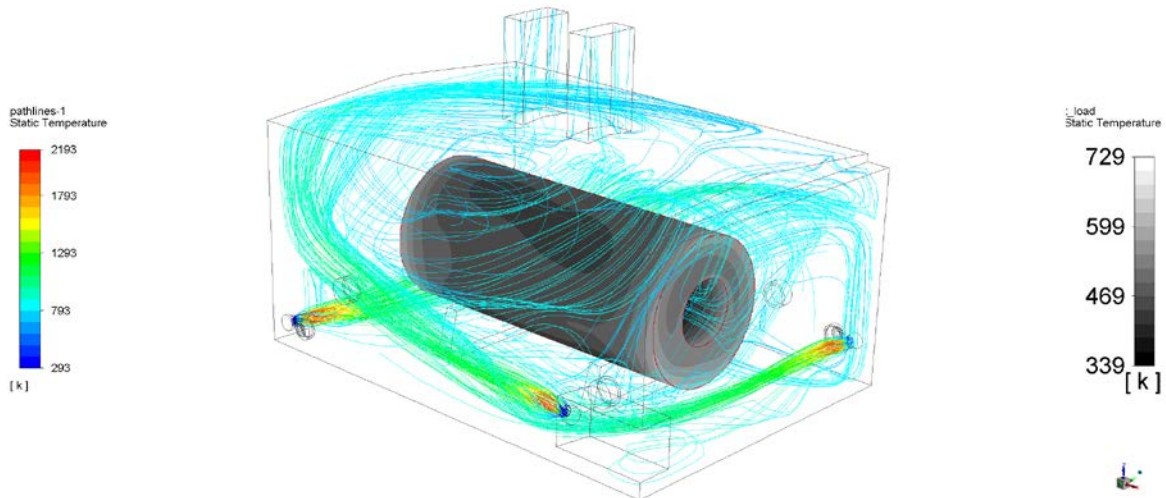

**Figure A3.** Gas path lines and load surface temperature after 2 h of the heat treatment—case 3.

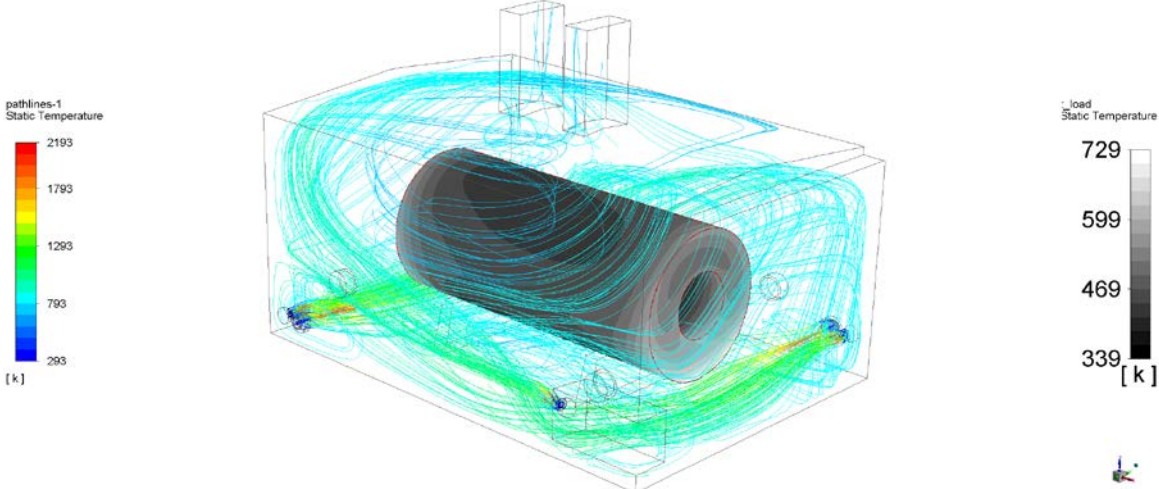

**Figure A4.** Gas path lines and load surface temperature after 2 h of the heat treatment—case 4.

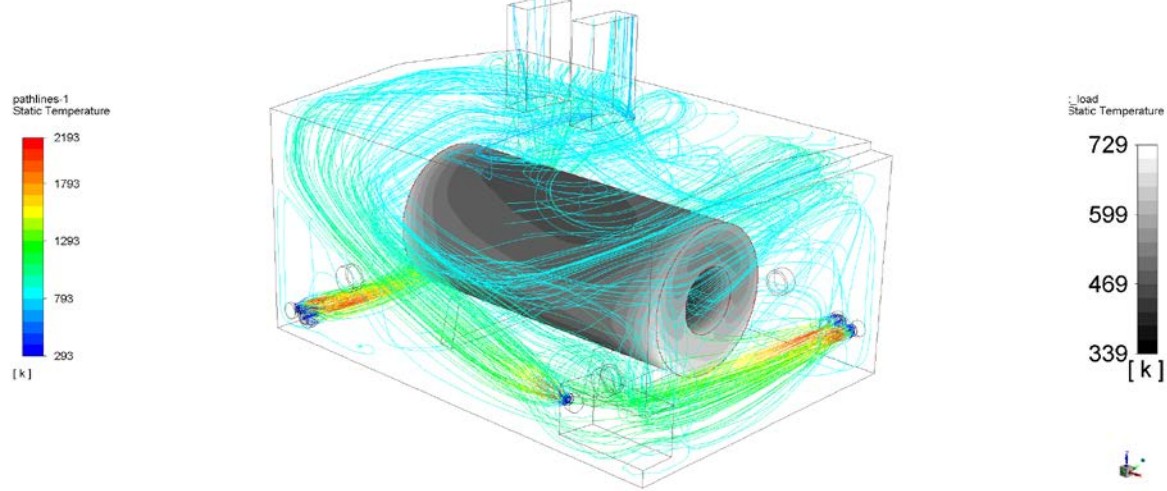

**Figure A5.** Gas path lines and load surface temperature after 2 h of the heat treatment—case 5.

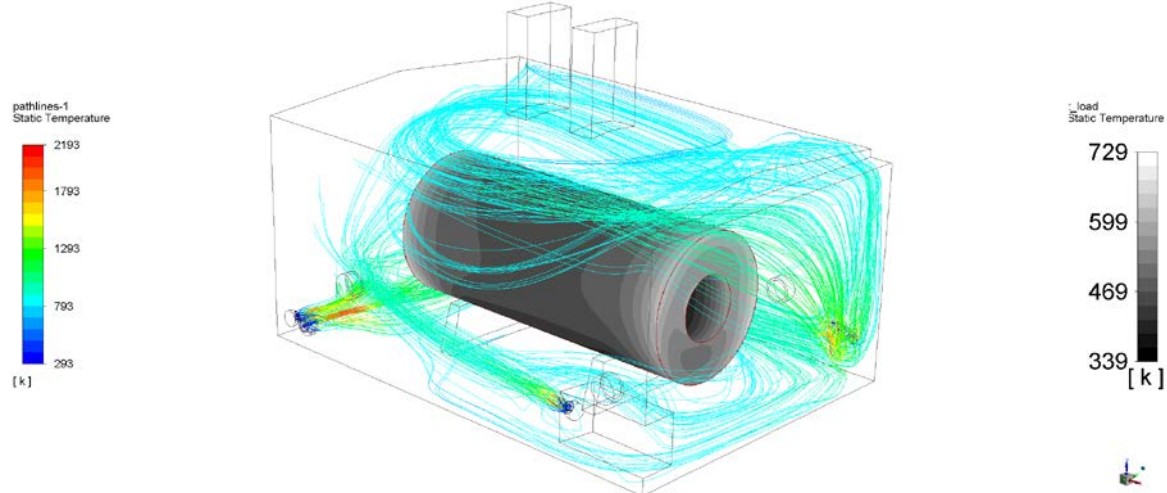

**Figure A6.** Gas path lines and load surface temperature after 2 h of the heat treatment—case 6.

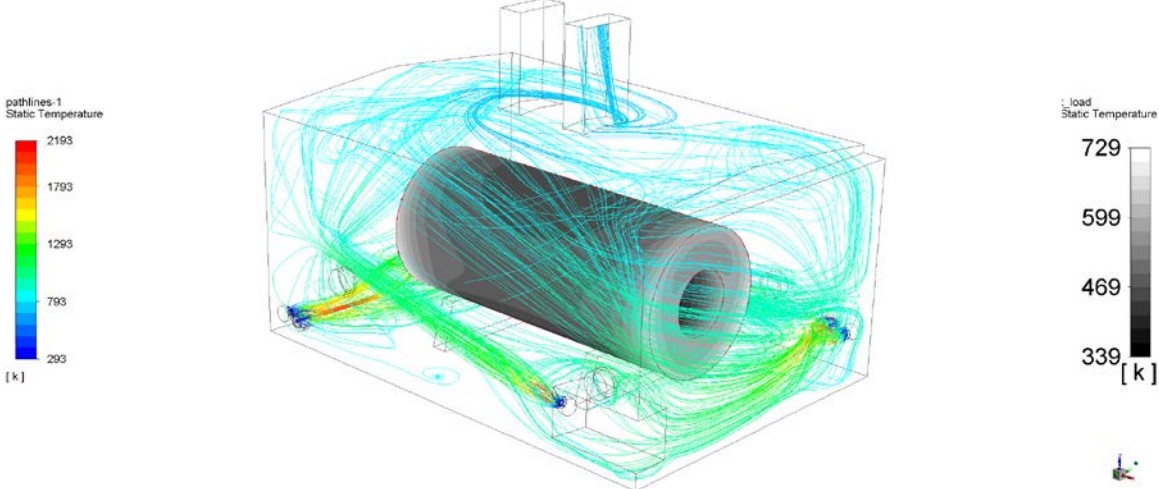

**Figure A7.** Gas path lines and load surface temperature after 2 h of the heat treatment—case 7.

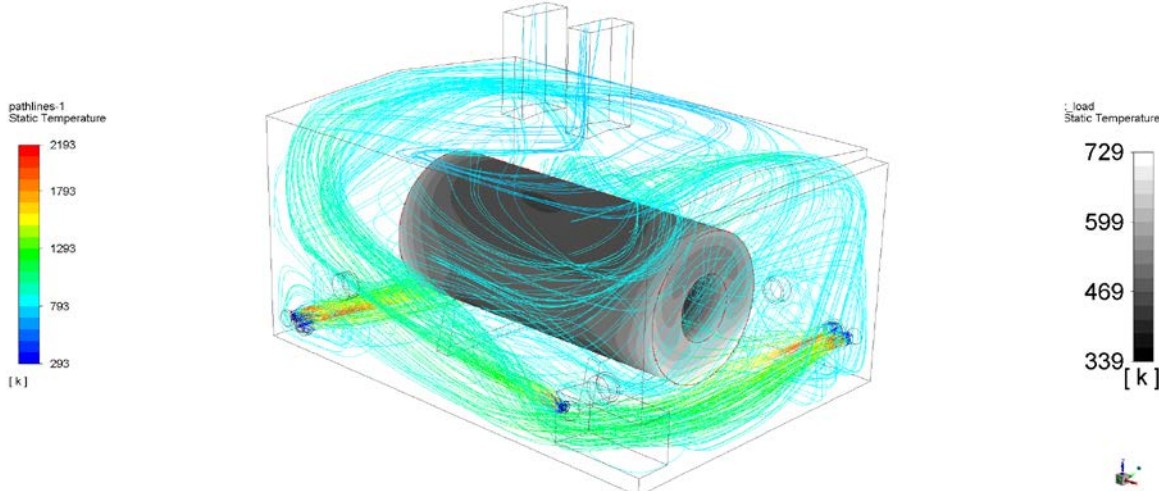

**Figure A8.** Gas path lines and load surface temperature after 2 h of the heat treatment—case 8.

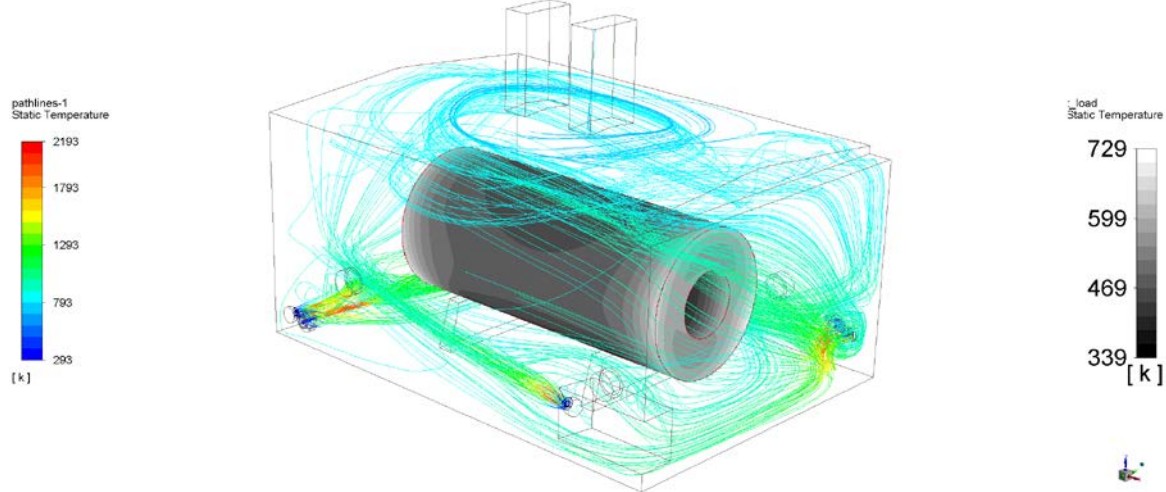

**Figure A9.** Gas path lines and load surface temperature after 2 h of the heat treatment—case 9.

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
