# Peer review of "Thermal Effects of Natural Gas and Syngas Co-Firing System on Heat Treatment Process in the Preheating Furnace"

_energies, doi:10.3390/en13071698_

Round 1
Reviewer 1 Report
This paper investigates numerically the effects of partial substitution of natural gas with biomass-derived synthesis gas on temperature of a typical load treated in an exemplary steel sector-preheating furnace. The results are to be compared with the ones gathered from the furnace fired in standard operation mode (leaning on natural gas only) in order to assess the possibility to apply this method of carbon dioxide emission reduction to other industrial units without worsening thermal parameters of the process. Overall, this paper showcases a successful parametric study on a preheating furnace. However, the discussions are lacking in that they focus only on the numerical values of the performance improvement/deterioration rather than giving insight into physics-based reasons why a given design variable would cause the resulting performance improvement/deterioration. In addition, the paper lacks of significant details, which made it not acceptable in its current form. The paper has good and interesting results; however, it can be accepted with major revision based on the following comments:
- Abstract section should be rewritten as the main results and conclusions should also be clearly stated at the end of the abstract.
- How did the authors’ achieve the grid independence test and where are its results? More details are needed.
- There is No information on the code validation and the authors have to compare their results with any available data from the literature.
- The accuracy of the numerical model need to be written.
- The results generally need to be discussed or even interpreted further. It is very essential to discuss what happens physically in the furnace?
- The paper need to be rechecked as there are several typos and grammatical errors. These should be corrected.
Author Response
Dear Reviewer,
we want to thank you for reviewing our paper and providing valuable remarks. We deeply appreciate your insight – with your input it was possible to increase the quality and clarity of this article.
Please see the attachment.
On behalf of the authors,
Piotr Jozwiak

Reviewer 2 Report
please see attached.

Author Response

(The authors gave the same response as above.)

Round 2
Reviewer 1 Report
The authors have to add their response on Point 2 "Point 2: How did the authors’ achieve the grid independence test and where are its results? More details are needed" in the revised manuscript to make the paper more valuable. Its very essential to show the grids and its independence test results in the revised manuscript.
Author Response
Dear Reviewer,
thank you again for reviewing our paper and providing valuable remarks.
Please see the attachment.
On behalf of the authors,
Piotr Jozwiak

Reviewer 2 Report
The reviewer’s comments are not fully addressed in a point-to-point manner.
The authors need to make change to the manuscript accordingly, rather than respond to the reviewer only. Besides, modifications to the manuscript are not clearly marked.
It is recommended to re-visit the revision before publication.
Author Response

(The authors gave the same response as above.)
